# Examining the Veterinary Electronic Antimicrobial Prescriptions for Dogs and Cats in the Campania Region, Italy: Corrective Strategies Are Imperative

**DOI:** 10.3390/ani13182869

**Published:** 2023-09-09

**Authors:** Valentina Foglia Manzillo, Maria Francesca Peruzy, Manuela Gizzarelli, Berardino Izzo, Paolo Sarnelli, Antonio Carrella, Giuseppina Vinciguerra, Claudia Chirollo, Nour El Houda Ben Fayala, Ines Balestrino, Gaetano Oliva

**Affiliations:** 1Department of Veterinary Medicine and Animal Productions, University of Naples Federico II, 80137 Naples, Italy; valentina.fogliamanzillo@unina.it (V.F.M.); mariafrancesca.peruzy@unina.it (M.F.P.); manuela.gizzarelli@unina.it (M.G.); nourelhouda.benfayala@unina.it (N.E.H.B.F.); gaetano.oliva@unina.it (G.O.); 2Veterinary Pharmacovigilance Center of Campania Region, 80137 Naples, Italy; berardino.izzo@regione.campania.it (B.I.); paolo.sarnelli@regione.campania.it (P.S.); info@farmacovigilanzavetcampania.it (A.C.); farmacovigilanza@unina.it (G.V.); c.chirollo@gmail.com (C.C.)

**Keywords:** antimicrobials, companion animals, veterinary electronic prescription, antimicrobial resistance

## Abstract

**Simple Summary:**

Companion animals are increasingly being recognised as important contributors to the spread of antimicrobial-resistant bacteria. The present work aimed to measure the antimicrobial drug prescribing in dogs and cats in the Campania Region, Italy by analysing the Veterinary Electronic Prescriptions (VEPs) between 2019 and 2020. The overall percentage of antibiotics prescribed in dogs was 65% and in cats 31.4%. In dogs, 90.5% of VEPs prescribed for systemic treatment included an antimicrobial Critically Important or Highly Important or Important for human medicine. The antimicrobials prescribed were mainly metronidazole–spiramycin (29.7%), amoxicillin–clavulanic (19.6%), enrofloxacin and cephalexin in dogs (16.5%) and enrofloxacin (22.6%) and amoxicillin–clavulanic acid (21.4%) in cats. Based on the results, the widespread use of broad-spectrum antimicrobials and the use of molecules for which limitations should be observed according to the EMA guidelines has emerged.

**Abstract:**

Companion animals are increasingly being recognised as important contributors to the spread of antimicrobial-resistant bacteria. The present work aimed to measure the antimicrobial drug prescribing in dogs and cats in the Campania Region, Italy by analysing the Veterinary Electronic Prescriptions (VEPs) between 2019 and 2020. The medical records associated with antimicrobial drug prescriptions were collected according to the drug administration (systemic or topical) and the rationale for the treatment chosen. In the period under investigation, 166,879 drugs were prescribed of which 129,116 (73.4%) were antimicrobial. A total of 83,965 (65%) antibiotics were prescribed to dogs, 40,477 (31.4%) to cats, and 4674 (3.6%) to other companion animals. In dogs, 90.5% of VEPs prescribed for systemic treatment included an antimicrobial Critically Important or Highly Important or Important for human medicine (WHO, 2018). The most widely prescribed class was fluoroquinolones. The antimicrobials prescribed were mainly metronidazole–spiramycin (29.7%), amoxicillin–clavulanic (19.6%), enrofloxacin and cephalexin in dogs (16.5%) and enrofloxacin (22.6%) and amoxicillin–clavulanic acid (21.4%) in cats. Based on the results, the widespread use of broad-spectrum antimicrobials and the use of molecules for which limitations should be observed according to the EMA guidelines has emerged.

## 1. Introduction

In recent years, antimicrobial resistance, which means “the inability or reduced ability of an antimicrobial agent to inhibit the growth of a bacterium” represents one of the most important worldwide threats to human and animal health [1]. Drug resistance is a serious public health concern that threatens to undermine decades of medical progress because, over the last years, the discovery of novel antibiotics for humans and animals has slowed while the use of antibiotics to treat bacterial infection has increased [2].

The overuse and misuse of antibiotics in humans and food-producing animals are considered to be the main reasons for the worldwide increase in antibiotic resistance in bacteria [3]. Food-producing animals play an important role in the transmission of antibiotic-resistant bacteria to humans and therefore the surveillance of antimicrobial use in most countries has been focused on them while little emphasis has been laid on the surveillance of antimicrobial use in pets [4,5]. Companion animals are increasingly being recognised as important contributors to the spread of resistant bacteria [6] and therefore with the recent EU Reg. 2019/6 on veterinary medicinal products, they have been included in the surveillance. However, in Europe, the surveillance intended specifically for companion animals will not be applied until the beginning of January 2029 [7].

The presence of resistant bacteria in pets may be due to their close relationship with humans, which may lead to a bacterial exchange between them or to the excessive or unnecessary use of antibiotics to treat infectious diseases or even non-infectious conditions (EMA Committee for Medicinal Products for Veterinary Use -CVMP-, 2018). Veterinarians, like other physicians, through correct and non-excessive antibiotic prescription, play a pivotal role in the fight against antibiotic resistance [6]. Veterinarians should prescribe antibiotic therapies only to treat infectious diseases and should choose the antibiotic according to the target species and the pathology [2]. Since 2019, veterinarians have been encouraged to prescribe antibiotics according to the European Medicines Agency’s scientific advice (EMA scientific advice) [8] in which the molecules are ranked into four groups according to both the risk that their use in animals causes to public health through the possible development of antimicrobial resistance and the need to use them in veterinary medicine.

However, off-label use of antimicrobials in dogs and cats, including antimicrobial drugs for humans, is common practice although there is still little scientific evidence supporting this [9]. Of particular concern is the prescription and the use of critically important antimicrobials (fluoroquinolones, third-generation cephalosporins, aminoglycosides, and carbapenems) in companion animals which may represent a significant risk for human health [5]. The continuous collection and analysis of data on the use of antimicrobials are essential to identify and implement interventions to prevent antimicrobial resistance in human and animal health [5]. To date, little data on antimicrobial use in dogs and cats are available in Italy. Smaller scale studies described patterns of antibiotic use based on data extracted from the clinical or prescribing records of veterinary hospitals and clinics [10,11]. These methods produce local insights into antibiotic practices but the use of teaching hospitals, in some cases, limits the generalisability of their findings.

The 2017 European Law has introduced electronic prescriptions for veterinary drugs and medicated feed stuff to control the distribution and administration of veterinary medicine along with the monitoring of antimicrobial resistance in bacteria. In Italy, the national information system for the management of the electronic prescription of veterinary medicine has been developed by the general direction of Animal Health and Veterinary Medicine (AHVM) of the Italian Ministry of Health in collaboration with the Experimental Zooprophylactic Institute of Abruzzo and Molise. The system is part of a broader simplification and digitalisation project by the Italian government, the 2015–2017 Simplification Agenda, which, for the topics related to veterinary health and food safety, provides for the introduction of innovative solutions to facilitate both the National Health Service and citizens in the fulfilment of regulatory obligations through so-called “dematerialisation”. A Veterinary Electronic Prescription (VEP) has been mandatory in Italy since 16 April 2019 and represents an important modification to the previous paper-based operating model for the management and traceability of veterinary medicines (Legislative Decree No. 193/06 implementing an EU Directive of 2004).

The present work aimed to measure the antimicrobial drugs prescribed in veterinary practice in dogs and cats in the Campania Region, Southern Italy by analysing the VEPs between April 2019 and December 2020 according to the drug administration (systemic or topical) and the rationale for the treatment chosen.

## 2. Materials and Methods

### 2.1. Pharma-Surveillance Information System—Vet Info

In this retrospective study, VEPs were collected from “VET INFO”, a Ministry of Health internet portal, where each veterinarian enters data through a personal account profile and issues a receipt (www.vetinfo.sanita.it, accessed on 6 September 2023). The system is divided into different types of prescriptions: companion animals including horses (not destined for human consumption), food-producing animals, and veterinary drugs from farm-held stock supply or veterinary stock medicines.

### 2.2. Data Collection

Only records of dogs and cats associated with antimicrobial drug prescriptions in the Campania Region, Southern Italy, between 16 April 2019 and 31 December 2020 were included in the analysis. 

The medical records of each dog and cat associated with antimicrobial drug prescription were collected according to the drug administration (systemic or topical) and the rationale for the treatment chosen: skin, respiratory, gastrointestinal, genito-urinary, mammary, metabolic, cardiovascular, neurological, oncological, orthopaedic, parasitic, ear and eye diseases, sepsis and general surgery. Data that did not belong to these categories were classified as “other”. No additional information regarding the group referred to as “other” could be extracted from the portal.

### 2.3. Data Management and Statistical Analysis

The collected data were recorded on spreadsheet software (Microsoft^®^ Excel^®^ 2018) and the differences in the frequency of antibiotic prescriptions were assessed using the chi-square test according to the recorded data (animal species and rationale for treatment). A probability value of less than 0.05 (*p* < 0.05) was defined as statistically significant.

## 3. Results

In the period under investigation, a total of 166,879 drugs were prescribed through the VEPs to companion animals, of which 129,116 (73.4%) were antimicrobial. A total of 83,965 (65%) antibiotics were prescribed to dogs, 40,477 (31.4%) to cats, and 4674 (3.6%) to other companion animals not included in the present analysis.

### 3.1. Antimicrobial Agents in the Canine Specie

In dogs, 62,941 (75%) and 21,024 (25%) antibiotic prescriptions and a total of 25 and 20 different molecules or combinations were prescribed for systemic and topical therapies, respectively (Table 1 and Table 2).

For systemic therapies, 31,505 (50%) VEPs contained one single active compound while 31,436 (50%) combined two molecules. The most widely prescribed antimicrobial class was beta-lactams (49%) (Figure 1). 

A total of 56,951 VEPs (90.5%) included an antimicrobial classified by WHO (2018) either as Critically Important or Highly Important or Important for human medicine (Table 1). According to the categorisation of antibiotics in the European Union published by the European Medicines Agency (11) most of the antibiotics prescribed (36.8%) belonged to the Category C: “Caution” (Figure 2).

In general, the most commonly prescribed drug was metronidazole–spiramycin (*n* = 18,699, 29.7%) followed by amoxicillin–clavulanic acid (*n* = 12,324, 19.6%), enrofloxacin (*n* = 10,399, 16.5%) and cephalexin (*n* = 6431, 10.2%) (Table 1).

Skin (*n* = 7531, 11%) and gastrointestinal disease (*n* = 6343, 10.1%) were the most frequent reasons for the antimicrobial therapeutic prescription. For skin diseases, the most common drugs prescribed were amoxicillin–clavulanic acid (*n* = 1834, 24.4%) (*p* < 0.5), cephalexin (*n* = 1482, 19.7%) (*p* < 0.05), metronidazole–spiramycin (*n* = 1421, 18.9%) (*p* < 0.05) and enrofloxacin (*n* = 1004, 13.3%) (*p* < 0.05). For gastrointestinal disease, metronidazole–spiramycin (*n* = 5128, 80.9%) (*p* < 0.05) was the most widely used (Table 1).

Metronidazole–spiramycin was also commonly prescribed for sepsis (*n* = 388, 27.8%), surgery (*n* = 17, 35.4%), cardiovascular disease (*n* = 14, 32.6%) and parasitic diseases (*n* = 169, 27.3%) (Table 1).

Amoxicillin–clavulanic acid was commonly prescribed to treat respiratory diseases (*n* = 815, 27.8%) (*p* < 0.05), mammary diseases (*n* = 107, 36.4%) (*p* < 0.05), and oncological diseases (*n* = 37, 33%) (*p* < 0.05). Doxycycline was commonly prescribed for ophthalmology diseases (*n* = 159, 48.9%) (*p* < 0.05), and metabolic disease (*n* = 114, 29.9%).

Enrofloxacin (*n* = 1866, 54.4%) was prescribed for genito-urinary diseases (*p* < 0.05), cephalexin (*n* = 265, 24.2%) for orthopaedic disorders (*p* < 0.05), and clindamycin (*n* = 111, 25.5%) and cephalexin (*n* = 105, 24.1%) for neurological diseases (Table 1).

For topical therapies, the combination of hydrocortisone aceponate/miconazole nitrate/gentamicin sulphate was the most prescribed (*n* = 4263, 20.3%) followed by terbinafine/florfenicol/betamethasone (*n* = 3460, 16.5%) and miconazole nitrate/polymyxin B sulphate/prednisolone acetate (*n* = 3428, 16.3%) (Table 2). 

Ear diseases (*n* = 18,317, 87.1%) were the most frequent reason for antimicrobial topical therapeutic prescriptions (*p* < 0.05), and hydrocortisone aceponate/miconazole nitrate/gentamicin sulphate was the most prescribed (*n* = 4263, 23.3%) (*p* < 0.05), followed by terbinafine/florfenicol/betamethasone (*n* = 3460, 18.9%). Miconazole nitrate/polymyxin B sulphate/prednisolone acetate (*n* = 1321, 71.9%) (*p* < 0.05) and tobramycin (*n* = 631, 72.6%) (*p* < 0.05) were the most frequently prescribed for skin and eye diseases, respectively (Table 2).

### 3.2. Antimicrobial Agents in the Feline Specie

In cats, 37,644 (93%) and 2833 (7%) antibiotic prescriptions and a total of 23 and 19 different molecules or combinations were prescribed for systemic and topical therapies, respectively (Table 3 and Table 4).

For systemic therapies, 24,687 (65.6%) VEPs contained one single active compound. The most widely prescribed antimicrobial class was fluoroquinolones (37.9%). 

In general, the most frequently prescribed antibiotics for systemic therapies were enrofloxacin (*n* = 8863, 23.5%) and amoxicillin–clavulanic acid (*n* = 8044, 21.4%) (*p* < 0.05) (Table 3).

A total of 26,270 VEPs (69.8%) included an antimicrobial classified by WHO (2018) as either Critically Important or Highly Important or Important for human medicine (Table 3). According to the categorisation of antibiotics in the European Union published by the European Medicines Agency (11), most of the antibiotics prescribed (80.5%) belonged to Categories B: “Restrict” and C: “Caution” (Figure 2).

Respiratory disease (*n* = 4544, 12.1%) followed by skin disease (*n* = 3300, 8.8%) was the most common reason for drug prescriptions. Doxycycline (*n* = 1376, 30.3%) and amoxicillin–clavulanic acid (*n* = 1158, 25.5%) were the most prescribed drugs for respiratory diseases (*p* < 0.05) while amoxicillin–clavulanic acid (*n* = 996, 30.2%) and enrofloxacin (*n* = 747, 22.6%) were for skin diseases (*p* < 0.05) (Table 3).

Metronidazole–spiramycin (*n* = 1356, 53%) (*p* < 0.05), enrofloxacin (*n* = 1832, 57.7%) (*p* < 0.05), clindamycin (*n* = 53, 53.5%) (*p* < 0.05), and sulfamethopyrazine (*n* = 297, 49.8%) (*p* < 0.05) were commonly prescribed for gastrointestinal, genito-urinary, neurology and parasitic diseases, respectively (Table 3). 

Doxycycline was commonly prescribed for ophthalmology (*n* = 96, 40.3%) (*p* < 0.05), metabolic (*n* = 95, 40.3%) (*p* < 0.05), sepsis (*n* = 247, 26.6%) (*p* < 0.05) and cardiovascular (*n* = 6, 66.7%) diseases (Table 3).

Amoxicillin–clavulanic acid (*n* = 107, 24.4%) and clindamycin (*n* = 84, 19.2%) (*p* < 0.05) were the molecules most prescribed for orthopaedic disorders, amoxicillin–clavulanic acid (*n* = 21, 29.6%) and metronidazole–spiramycin (*n* = 14, 19.7%) for mammary diseases, enrofloxacin (*n* = 6, 37.5%) and amoxicillin–clavulanic acid (*n* = 5, 31.3%) for general surgery and amoxicillin–clavulanic acid (*n* = 18, 34%) and amoxicillin (*n* = 14, 26.4%) for oncological diseases (*p* < 0.05). 

For topical therapies, a combination of miconazole nitrate/polymyxin B sulphate/prednisolone acetate (*n* = 1081, 38.2%) (*p* < 0.05) and thiabendazole/neomycin/dexamethasone (*n* = 739, 26.1%) (*p* < 0.05) were the most prescribed (Table 4).

Ear disease was the most common reason for the antimicrobial therapeutic prescription (*n* = 1675, 59.1%) (*p* < 0.05) and miconazole nitrate/polymyxin B sulphate/prednisolone acetate (*n* = 791, 47.2%) (*p* < 0.05) was the most prescribed (Table 4). 

Thiabendazole/neomycin/dexamethasone (*n* = 402, 53.3%) (*p* < 0.05) and tobramycin (*n* = 324, 80.2%) (*p* < 0.05) were commonly prescribed for skin and eye diseases, respectively (Table 4).

## 4. Discussion

In the present study, a total of 129,116 antibiotics were prescribed to companion animals between 2019 and 2020 in the Campania Region. Over this period, the percentage of antimicrobials prescribed out of the total drugs prescribed (73.4%) was higher than those recorded at the University Veterinary Teaching Hospital (OVUD) in our previous research (41.6%) [11], compared to those recorded previously in another Italian study conducted at the hospital of the University of Pisa (30.6%) [12] and those reported by Schnepf et al. [7] at a veterinary teaching hospital in Germany (17.8%). Although it is speculative, based on the results of the present work, the trend of prescribing antimicrobials is likely to be higher in private veterinary practices. In contrast with the studies of Chirollo et al. [11], and Schnepf et al. [7], in which only animals referred to hospitals were included in the analysis, in the present work, VEPs of all practitioners working in the Campania Region were recorded, thus providing a broader picture of antimicrobial prescription practices in companion animals. In general, the reason for the high number of antimicrobials prescribed by practitioners could be a fear of complications or owner dissatisfaction [13]

A higher number of antimicrobial prescriptions was found in dogs (*n* = 83,965, 65%) compared with cats (*n* = 40,477, 31.4%). Results are in line with those recorded by Hardefeldt et al. [14] and Hur et al. [15], which reported a higher use of antimicrobials in dogs. The higher number of antimicrobial prescriptions found in dogs could be explained by the higher number of routine preventative health examinations performed on dogs compared to cats [15].

However, in the study of Escher et al. [12] a significantly higher percentage of antimicrobial prescriptions was recorded for cats (cats: 44% vs. dogs: 27.3%) and in the studies of Murphy et al. [16] and Buckland et al. [17] a similar percentage of antimicrobial prescription was observed in dogs and cats.

In general, most VEPs contained one single-active compound. Fluoroquinolones were commonly prescribed both in dogs and in cats. The use of fluoroquinolones in dogs and cats was much more common in the present study than in any other study available in the other literature on therapies [10,16,18].

Furthermore, our study demonstrated that there was widespread use of antimicrobials classified by EMA (2019) as critically important for human health (Table 1 and Table 3). Unfortunately, the prescription of these antibiotics is common in small veterinary practices [19]. This critical behaviour is of particular concern for the risk of the emergence and transmission of bacteria resistant to antimicrobials that are considered to be of the greatest importance for human medicine. According to EMA [12], only the antibiotics included in category D should be used in veterinary medicine as a first-line treatment for animal infections. In the present work, the number of prescriptions containing antibiotics grouped into this latter category was lower compared to those prescribing antibiotics grouped into categories B and C. Results are of particular concern because category B includes antibiotics critically important for human medicine and therefore the use of them in veterinary medicine should be limited.

In general, the most prescribed drugs were metronidazole–spiramycin, amoxicillin–clavulanic, enrofloxacin and cephalexin in dogs and enrofloxacin and amoxicillin–clavulanic acid in cats. Results of the present work are in line with those reported in our previous research [11] in which cefalexin (18%), amoxicillin/clavulanate (18%), and metronidazole–spiramycin (10%) were commonly prescribed in dogs [11]. However, in that previous study, the association of cephalexin and clindamycin, which was not recorded in the present work, was also highly prescribed (17%). Regarding cats, amoxicillin–clavulanic acid (41%) and enrofloxacin (17%) were also very frequently recorded in the research of Chirollo et al. [11]. The third-generation cephalosporin cefovecin, widely used in the UK and Belgium in cats was not commonly prescribed in the present work [17,18,20].

In addition, it turned out that metronidazole was used in combination with spiramycin; antimicrobial associations (such as metronidazole–spiramycin) are used in specific cases to obtain a synergetic effect, to allow lower doses of either the active ingredient or to avoid the emergence of resistance [21]. In the present research, it was the most frequently prescribed antibiotic in dogs for gastrointestinal, sepsis, cardiovascular, and parasitic diseases and general surgery, and in cats for gastrointestinal disease. The results are of particular concern since metronidazole alone or in combination with spiramycin can produce severe side effects in cats and dogs, therefore prudent use is essential [7].

Potentiated penicillins (amoxicillin–clavulanic acid) and first-generation cephalosporins (such as cephalexin) are used in veterinary medicine as a first-line therapy. In the present research, amoxicillin–clavulanic acid was the most frequently prescribed antibiotic in dogs for skin, respiratory, mammary, and oncological diseases and in cats for skin, mammary and oncological diseases and orthopaedic disorders. Amoxicillin–clavulanic acid was also the most commonly used potentiated agent for both species (dogs: 44.7%; cats: 29%) in the UK [17]. It is a broad-spectrum and inexpensive antimicrobial, and it is often used for a suspected infection without a culture check or antibiogram. Cephalexin, commonly prescribed here during orthopaedic disorders in dogs is generally used as a relatively narrow-spectrum antimicrobial under current guidelines [11]. However, in the study of Chirollo et al. [11], the association of cefalexin and clindamycin was preferred for orthopaedic diseases. 

Interestingly, the use of enrofloxacin, which should be used in dogs and cats as a second-line therapy, was much more common in our research than in any other study available in the literature [5,10,17] As Lhermie et al. reported [19], [ fluoroquinolones are used to treat urinary tract infections. In the present study, fluoroquinolones enrofloxacin was commonly used for genito-urinary diseases both in dogs and cats.

Skin diseases were one of the most common reasons for antimicrobial treatment in both dogs and cats. These results are in line with those reported by Mouiche et al. [5]. In the present work, amoxicillin–clavulanic acid was the most frequently prescribed antimicrobial. In cats, the results are in line with those reported by Murphy et al. [16]. They are, however, in contrast with those reported for dogs by Escher et al. [12] and Murphy et al. [16], in which cephalosporins were the most commonly prescribed antibiotics in the case of skin diseases. Skin diseases are among the most common consultation reasons in small-animal practice and the use of topical medications instead of systemic medications should increase [22].

Even if the national guidelines promote the use of topical rather than systemic antimicrobials where appropriate, in the present study, a preferential use of systemic therapies emerged. In general, the use of systemic treatments increases the exposure of the gut microbial population to antimicrobials and therefore increases the risk of the occurrence of antibiotic-resistant bacteria [22].

Ear diseases were, however, the most common reason for topical antimicrobial therapeutic prescriptions in both dogs and cats. Different bacteria, such as *Pseudomonas*, *Proteus*, *Enterococcus*, *Streptococcus*, and *Corynebacterium* can cause ear infections. Acute and uncomplicated otitis externa can often be treated successfully with antibiotics and topical therapies and is typically preferred [23].

## 5. Conclusions

In the present work, VEPs of all practitioners working in the Campania Region were recorded and analysed, and this provided a comprehensive picture of antimicrobial prescription practices in companion animals. The main antimicrobials that were prescribed during the study period were metronidazole–spiramycin, amoxicillin–clavulanic, enrofloxacin and cephalexin in dogs, and enrofloxacin and amoxicillin–clavulanic acid in cats. Based on the results, a widespread use of broad-spectrum antimicrobials has emerged. Antimicrobials are important for animal welfare but need to be used prudently. In the present work, in order to manage different infections, veterinarians mostly turned to molecules for which limitations should be observed instead of prescribing first-line antibiotics. Moreover, further efforts must be made to decrease the overall use of systemic antibiotics in companion animals. This evidence could be used by governing bodies to develop actions for more stringent controls of the use of antimicrobials in veterinary practice and could be used for future informative campaigns on the correct use of antimicrobials.

However, in the present study, the doses and quantity of active substances prescribed by practitioners were not evaluated. Further research that includes these latter aspects along with the actual use of antibiotics by pet owners will be performed in the future. 

## Figures and Tables

**Figure 1 animals-13-02869-f001:**
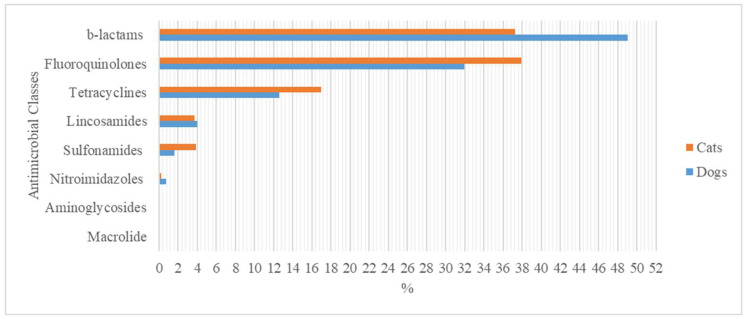
Compounds prescribed for systemic therapies in dogs and cats grouped according to the pharmacological class.

**Figure 2 animals-13-02869-f002:**
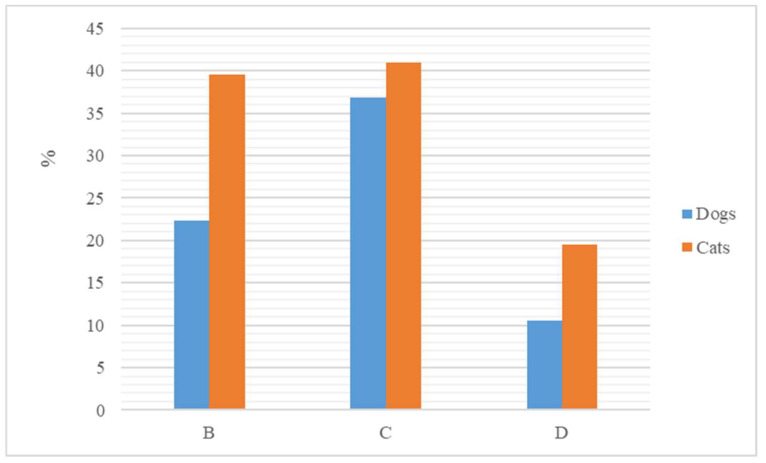
Percentage of antibiotics prescribed to dogs and cats grouped according to the categorisation of antibiotics in the European Union published by the European Medicines Agency (EMA) as B (category B, “Restrict”), C (category C, “Caution”) and D (category D, “Prudence”).

**Table 1 animals-13-02869-t001:** Antibiotics prescribed in dogs for systemic therapies to treat the following diseases: skin (SK), respiratory (RES), gastrointestinal (GI), genito-urinary (GU), ophthalmology (OPH), orthopaedic (ORT), mammary (MAM), sepsis (SEP), general surgery (SUR), metabolic (MET), cardiovascular (CAR), neurological (NEU), oncological (ONC) and parasitic (PAR). Data that did not belong to these categories were classified as “other”.

Antibiotics	Rationale for the Treatment Chosen															
	SK	RES	GI	GU	OPH	ORT	MAM	SEP	SUR	MET	CAR	NEU	ONC	PAR	Other	Tot.
Amikacin *^C^								1				1			7	9
Amoxicillin *^D^	9	5	2	5		3	1	4		1					68	98
Amoxicillin–Clavulanic acid *^C^	1834	815	352	556	57	179	107	181	15	55	13	49	37	6	8068	12,324
Ampicillin *^D^															9	9
Benzylpenicillin *^D^	1							1								2
Benzylpenicillin–Dihydrostreptomycin *	38	20	8	8		3	5	16	1	2		1		1	95	198
Cefadroxil **^C^	714	166	31	95	12	76	30	45	1	9		35	11	1	1387	2613
Cefalexin **^C^	1482	213	90	149	16	265	32	82	11	10	2	105	6	7	3961	6431
Cefovecin **^B^	3		1												8	12
Chlortetracycline **^D^	6		16		1			1		1				86	104	215
Clindamycin **^C^	231	24	37	17	5	220	4	80		8		111	7	5	1011	1760
Doxicicline ^D^	188	697	106	80	159	84	3	351		114	5	31	9	157	3260	5244
Enrofloxacin *^B^	1004	571	188	1866	26	104	36	175	2	59	9	37	23	16	6283	10,399
Formosulfathiazole **^D^	5						1								2	8
Gentamicin *^C^	1		1					1		1					6	10
Kanamycin *–Isopropamide Iodide			71							1					91	163
Lincomycin **–Spectinomycin ***	1			6		1		3				3			28	42
Marbofloxacin *^B^	325	105	35	416	31	45	9	48	1	6		48	4		1769	2842
Metronidazole ***^D^	3		104					1		2			1	11	209	331
Metronidazole ***–Spiramycin *	1421	247	5128	179	16	104	65	388	17	108	14	8	13	169	10,822	18,699
Oxytetraciclin^D^	1	4		3				6		2				1	39	56
Pradofloxacin *^B^	262	55	6	48	2	11	1	13				7	1		364	770
Sulphadiazine/Sulphadimethoxaxole–Trimetoprim															2	2
Sulfametopyrazine ^D^	2	4	166			1				2				159	354	688
Tylosin *^C^		2	1												13	16
Tot.	7531	2928	6343	3428	325	1096	294	1397	48	381	43	436	112	619	37,960	62,941

Antimicrobial classified by WHO (2018) as Critically Important *, Highly Important ** and Important *** for human medicine. Antimicrobial classified by EMA (2019) as B (category B, “Restrict”), C (category C, “Caution”) and D (category D, “Prudence”).

**Table 2 animals-13-02869-t002:** Antibiotics prescribed in dogs for topical therapies to treat skin, ear, and eye disease.

Antibiotics	Rationale for the Treatment Chosen			
	Skin Disease	Ear Disease	Eye Disease	Tot.
Betamethasone/Clotrimazole/Gentamicin		1182		1182
Cloramphenicol/Betamethasone			8	8
Clostebol/Paromomycin/Prednisolone	114			114
Diethanolamine Fusidate/Framycetin Sulphate/Nystatin/Prednisolone		218		218
Econazole/Flumetasone/Gentamicin/Tetracaine	104	237		341
Enrofloxacin/Silver Sulfadiazine		797		797
Fluocinolone/Neomycin			175	175
Fusidic Acid			16	16
Fusidic Acid/Betamethasone	288			288
Gentamicin			39	39
Hydrocortisone Aceponate/Miconazole Nitrate/Gentamicin Sulphate		4263		4263
Marbofloxacin/Clotrimazole/Dexamethasone		2058		2058
Marbofloxacin/Gentamicin Sulphate/Ketoconazole/Prednisolone		810		810
Marbofloxacin/Ketokonazole/Prednisolone	11			11
Miconazole Nitrate/Polymyxin B Sulphate/Prednisolone Acetate	1321	2107		3428
Orbifloxacin/Posaconazole/Mometasone Furoate		1322		1322
Rifaximin/Colistin/Miconazole/Carbarele/Triamcinolone		1230		1230
Terbinafine/Florfenicol/Betamethasone		3460		3460
Thiabendazole/Neomycin/Dexamethasone		633		633
Tobramycin			631	631
Tot.	1838	18,317	869	21,024

**Table 3 animals-13-02869-t003:** Antibiotics prescribed in cats for systemic therapies to treat the following diseases: skin (SK), respiratory (RES), gastrointestinal (GI), genito-urinary (GU), ophthalmology (OPH), orthopaedic (ORT), mammary (MAM), sepsis (SEP), general surgery (SUR), metabolic (MET), cardiovascular (CAR), neurological (NEU), oncological (ONC) and parasitic (PAR). Data that did not belong to these categories were classified as “other”.

Antibiotics	Rationale for the Treatment Chosen															
	SK	RES	GI	GU	OPH	ORT	MAM	SEP	SUR	MET	CAR	NEU	ONC	PAR	Other	Tot.
Amikacin *^C^			1												1	2
Amoxicillin *^D^	73	71	20	17	4	5	4	3		4		4	14	7	290	516
Amoxicillin–clavulanic Acid *^C^	996	1158	234	508	55	107	21	188	5	34	1	7	18	9	4703	8044
Benzylpenicillin–Dihydrostreptomycin *		9	2	2	1			5							39	60
Cefadroxil **^C^	235	106	16	58	2	9	5	17	2	9	1	1	1	3	462	927
Cefalexin **^C^	454	287	37	95	10	58	10	50	1	8		5	2	3	1669	2689
Cefovecin **^B^	2	1	1					1					1		21	27
Ceftiofur **^B^															1	1
Chlortetracycline **^D^	1	3	25											134	124	287
Clindamycin **^C^	133	44	19	5	6	84		58		7		53	5	10	801	1225
Doxycycline ^D^	90	1376	105	102	96	17	4	247		95	6	8	4	52	3049	5251
Enrofloxacin *^B^	747	825	203	1832	23	75	11	161	6	32	1	12	1	9	4925	8863
Formosulfathiazole **^D^	3														3	6
Gentamicin *^C^															2	2
Kanamycin *–Isopropamide iodide			95							1				1	88	185
Lincomycin **–Spectinomycin ***				5				1								6
Marbofloxacin *^B^	115	104	39	303	14	16	1	19		8		5	1		961	1586
Metronidazole ***^D^	1	1	18				1			1					42	64
Metronidazole ***–Spiramycin *	233	108	1356	24	8	29	14	109	2	34		3	5	70	2661	4656
Oxytetraciclin ^D^		1						2							15	18
Pradofloxacin *^B^	213	398	44	220	18	38		67		3		1	1	2	965	1970
Sulfametopyrazine	2	52	342	3	1			1						297	560	1258
Tylosin *^C^															1	1
Tot.	3300	4544	2557	3174	238	438	71	929	16	236	9	99	53	597	21,383	37,644

Antimicrobials classified by WHO (2018) as Critically Important *, Highly Important **, and Important *** for human medicine. Antimicrobials classified by EMA (2019) as B (category B, “Restrict”), C (category C, “Caution”) and D (category D, “Prudence”).

**Table 4 animals-13-02869-t004:** Antibiotics prescribed in cats for topical therapies to treat skin, ear, and eye diseases.

Antibiotics	Rationale for the Treatment Chosen			
	Skin Disease	Ear Disease	Eye Disease	Tot.
Betamethasone/Clotrimazole/Gentamicin		46		46
Clostebol/Paromomycin/Prednisolone	9			9
Diethanolamine Fusidate/Framycetin Sulphate/Nystatin/Prednisolone		33		33
Econazole/Flumetasone/Gentamicin/Tetracaine	24	42		66
Enrofloxacin/Silver Sulfadiazine		53		53
Fluocinolone/Neomycin			58	58
Fusidic Acid			3	3
Fusidic Acid/Betamethasone	27			27
Gentamicin			19	19
Hydrocortisone Aceponate/Miconazole Nitrate/Gentamicin Sulphate		65		65
Marbofloxacin/Clotrimazole/Dexamethasone		56		56
Marbofloxacin/Gentamicin Sulphate/Ketoconazole/Prednisolone		16		16
Marbofloxacin/Ketokonazole/Prednisolone	2			2
Miconazole Nitrate/Polymyxin B Sulphate/Prednisolone Acetate	290	791		1081
Orbifloxacin/Posaconazole/Mometasone Furoate		25		25
Rifaximin/Colistin/Miconazole/Carbarele/Triamcinolone		171		171
Terbinafine/Florfenicol/Betamethasone		40		40
Thiabendazole/Neomycin/Dexamethasone	402	337		739
Tobramycin			324	324
Tot.	754	1675	404	2833

## Data Availability

Publicly available datasets were analyzed in this study. This data can be found here: https://www.farmacovigilanzavetcampania.it/ (accessed on 6 September 2023).

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
