# Peer review of "Examining the Veterinary Electronic Antimicrobial Prescriptions for Dogs and Cats in the Campania Region, Italy: Corrective Strategies Are Imperative"

_animals, 2023, doi:10.3390/ani13182869_

Round 1
Reviewer 1 Report
In the present manuscript, the authors measured the prescription of antimicrobial drugs in veterinary practice in dogs and cats in the Campania region, Italy, by analyzing veterinary electronic prescriptions between April 2019 to December 2020.
I only have a few suggestions/questions:
Authors should review the p-values and remove them from the manuscript if they are not statistically significant (for example, lines 178, 179, 183, 187…).
What reasons could explain the higher number of antimicrobial prescriptions found in dogs compared with cats, as well as the higher percentage of antimicrobial critically important or highly important or important for human medicine prescribed for systemic treatment in dogs vs cats?
Authors should discuss the strengths and limitations of their manuscript.
Based on the results obtained, are you planning any type of intervention or study?
An article that could be consulted and cited by the authors:
Galarce, N.; Arriagada, G.; Sánchez, F.; Venegas, V.; Cornejo, J.; Lapierre, L. Antimicrobial Use in Companion Animals: Assessing Veterinarians’ Prescription Patterns through the First National Survey in Chile. Animals 2021, 11, 348. https://doi.org/10.3390/ani11020348
Minor comments:
Line 158: “Table 2” change to “Table 3”. In the same Table, correct “b-Lactams”.
Line 201: “3.3” change to “3.2”.
Line 206: “dogs” change to “cats”.
Lines 249, 250 and 252: remove “diseases”.
Line 261: “73.73%” change to “73.37%”.
Line 284: “Tables 1 and 5” change to “Tables 1 and 4”.
Line 329: “Mouiche et al. (2021)” change to “Mouiche et al. [5]”.
Moderate editing of English language required.
Author Response
Dear Reviewer,
We thank you for your valuable comments, we made the necessary adaptations and corrections throughout the manuscript (highlighted in yellow. Green corrections were made by native English speaker). A detailed answer on each remark is included in the rebuttal.

Reviewer 2 Report
A cross-sectional study of antimicrobial prescriptions for dogs and cats attending general practices in a region of Italy has been performed. The data source seems promising but there are many areas of the paper that need to be addressed before it is suitable for publication.
Materials and methods: I suggest you have a sub-heading that has more detailed information on the data source.
Are the medical records submitted with the prescription?
What methods did you use to classify the rationale for antimicrobial treatment? Who made this assessment?
How did you decide on the categories for rationale?
Collected data was initially recorded using excel - what did you use subsequently?
Results:
Please use 2 significant figures for percentages through-out.
Table 1 and 4 - more than 1/2 of the prescriptions are in the "other" category. This is concerning. Somewhere you need to give a complete list of the data - supplementary information perhaps.
Table 4 is mislabeled - should be antibiotics prescribed for cats
WHO recommends using regional importance ratings where they have been developed. If Italy doesn't have a system then the EMA categorisation should be used here.
What is the significance of single vs multiple active agents? I don't think these should be highlighted in the results.
Similarly, "antimicrobial associations" in table 2 should be further classified as they are nearly 1/2 the records.
Instead of reporting the total proportion of critically, highly and important please report proportions that are category A, B, C & D rather than grouping them all together.
Through-out the results it is not clear what you are testing with your chi square test. Please be very clear about this and use APA format for reporting test result.
Discussion:
Line 260- the study period was not 2 years
Line 262-265 - how confident are you on the completeness of the data? Is reporting predominately for antimicrobial use surveillance? If so, are vets reporting antimicrobials regularly but not reporting other prescriptions? How is the data audited?
Line 274 - the comparison is not appropriate. The Escher study looked at the proportion of cats that were administered an antibiotic not the proportion of prescriptions that were administered to cats. These are 2 very different measurements.
You should also consider:
Hardefeldt et.al. https://doi.org/10.1016/j.vetmic.2018.09.010
Hur et al. https://doi.org/10.1371/journal.pone.0230049
The conclusion is oversimplified. Consider category A antimicrobial use and what it is being used for and why? Further study is likely needed to investigate this further.
Editing for English language is required.
Author Response

(The authors gave the same response as above.)

Reviewer 3 Report
Dear authors,
Many thanks for your interesting paper. That is really good to read the analyses of such a database. Very promising for the future!
Please find below my comments and suggestions to improve your manuscript. I would also recommend to have a quality review from a native english speaker.
L19 double space?
L31 "xxx drugs/AB were prescribed": please rephrase this formulation. it refers to a number of prescription and not about a number of drugs. Please update all the manuscript accordingly
L46-60: i would shorten this part as there is only a small relation to the topic of the article. I would go directly into the topic.
L63: double space?
L98: use "since" instead of "from"
L101: prescribed instead of prescribing
L109: recipe is probably not what you meant
L117: drug delivery is not the right term for the application mode
L117: do not use "local treatment" but "topical treatment". Change in all the text.
L118: "genitourinary" does not exist. Please correct it in the whole manuscript
P107-112: can you please provide more information about the system? What kind of data do they share? is it mandatory? do they have to enter data manually? (linked to the quality of data, coverage...)
L122: "were initially", did you only Excel for all these analyses and the management of all data? Please mention any other tool you used
L124: please provide more information about your chi-square analyses. how and why did you apply/use it?
L128 please write the numbers in a more readable way using separators ex. 166'876
L129 what were the other prescriptions? are vaccines and antiparasitics also included in Vet Info?
L134-136 if I unterstand correctly you summarised the 2 years together. why didn't you have separated analyses for each year? would it be possible to see a trend?
L159: wrong numbering of the table
Tables: please specify that it refers to the number of prescription in the titles to make it clear for the reader
L153-155: please split the two ideas/results in two sentences for understanding purposes.
L166: names of AM should not have a capital letter. please adapt accordingly in the whole manuscript
L167 linked to the question about the chi-square analyses, please explain in details what this p-value means in the context of the result. What was compared? what does it bring in this paper?
L192 please specify that this paragraph refers to the topical AM treatments. in general I found it difficult to identify the paragraph and tables.
L252 "Disease" has a capital letter
L339: double space?
Discussion: I suggest to expand the discussion on the following topics
- explanations for the proportion of AM prescriptions. Do veterinarians have a profit from selling AM? did you compare/discuss the results with other provinces? do you have potentiel explanations?
- what is the quality of data? did you identify "wrong" prescriptions? how did you manage them? can a prescription be for several animals?
- why did you only study the number of prescription? do you have information about dosis, quantity of active substance?
- can you link your results to national guidelines?
- L298: can you please justify/explain your comparison of italian data with cameroonese data? i am not sure this reference is really relevant.
Dear authors,
Many thanks for your interesting paper. That is really good to read the analyses of such a database. Very promising for the future!
Please find below my comments and suggestions to improve your manuscript. I would also recommend to have a quality review from a native english speaker.
L19 double space?
L31 "xxx drugs/AB were prescribed": please rephrase this formulation. it refers to a number of prescription and not about a number of drugs. Please update all the manuscript accordingly
L46-60: i would shorten this part as there is only a small relation to the topic of the article. I would go directly into the topic.
L63: double space?
L98: use "since" instead of "from"
L101: prescribed instead of prescribing
L109: recipe is probably not what you meant
L117: drug delivery is not the right term for the application mode
L117: do not use "local treatment" but "topical treatment". Change in all the text.
L118: "genitourinary" does not exist. Please correct it in the whole manuscript
P107-112: can you please provide more information about the system? What kind of data do they share? is it mandatory? do they have to enter data manually? (linked to the quality of data, coverage...)
L122: "were initially", did you only Excel for all these analyses and the management of all data? Please mention any other tool you used
L124: please provide more information about your chi-square analyses. how and why did you apply/use it?
L128 please write the numbers in a more readable way using separators ex. 166'876
L129 what were the other prescriptions? are vaccines and antiparasitics also included in Vet Info?
L134-136 if I unterstand correctly you summarised the 2 years together. why didn't you have separated analyses for each year? would it be possible to see a trend?
L159: wrong numbering of the table
Tables: please specify that it refers to the number of prescription in the titles to make it clear for the reader
L153-155: please split the two ideas/results in two sentences for understanding purposes.
L166: names of AM should not have a capital letter. please adapt accordingly in the whole manuscript
L167 linked to the question about the chi-square analyses, please explain in details what this p-value means in the context of the result. What was compared? what does it bring in this paper?
L192 please specify that this paragraph refers to the topical AM treatments. in general I found it difficult to identify the paragraph and tables.
L252 "Disease" has a capital letter
L339: double space?
Discussion: I suggest to expand the discussion on the following topics
- explanations for the proportion of AM prescriptions. Do veterinarians have a profit from selling AM? did you compare/discuss the results with other provinces? do you have potentiel explanations?
- what is the quality of data? did you identify "wrong" prescriptions? how did you manage them? can a prescription be for several animals?
- why did you only study the number of prescription? do you have information about dosis, quantity of active substance?
- can you link your results to national guidelines?
- L298: can you please justify/explain your comparison of italian data with cameroonese data? i am not sure this reference is really relevant.
Author Response

(The authors gave the same response as above.)

Round 2
Reviewer 2 Report
Thank you for your revisions however I find the manuscript still requires substantial work before it is suitable for publication.
As with my previous review - please present percentages with only 2 significant figures 0 ie 60.05% should be documented as 60% and 9.35% should be documented as 9.4% etc. In addition, please use commas rather than apostrophes in large numbers - ie 83,965 not 83'965.
English language editing is still required through-out.
Abstract and simple summary:
Second line antibiotics is a term that is used in guidelines not in relation to spectrum. Some higher importance antimicrobials can be first line treatments for certain conditions according to guidelines.
Refer to EMA importance rating not WHO
Line 35 - delete "commonly"
Line 35 - I dispute this - the most widely prescribed class was beta lactase if you include amoxyclav. What is your rationale for excluding it?
Line 38/39 - this sentence doesn't make sense - please reword.
Introduction:
Line 46-50 - incomplete sentence
No need for AR, ARB and AMU acronyms - the use of these terms is minimal throughout the manuscript.
Line 78 - there are many published research papers looking at large scale AMU from companion animals - VetCompass Australia, VetCompass UK, SAVSNET, pet insurance data from Australia
Introduction needs to include an explanation of the EMA importance ratings.
Methods:
Line 106 - "recipe" seems like the wrong word - receipt?
Line 111: What data is collected by VETINFO? What are the fields in your data? In response to my last review, you indicated that the medical record is available for these prescriptions - if this is the case then you must be able to validate the data classification for "rationale" and further investigate the "other" category. Unless you understand how vets are using the system I have serious concerns over the validity of the data.
Data analysis - needs to be expanded and clarified. I still don't understand why and how you used the chi-squared test.
Results:
Table 1 and 3 - include proportions as well as numbers. Use EMA ratings not WHO.
Through-out - it is not clear what the chi-square test is reporting a difference in and APA format still not being used.
Line 158-159: I find this misleading - beta-lactams (when including amoxyclav) are the most commonly used - it doesn't make sense to me that you would exclude amoxyclav because of the clav component.
Line 169-170: delete analysis according to WHO classification - this is misleading and not appropriate. Use EMA classifications and present results for all classes not just category C. You have considerable focus on the use of the more important antibiotics in this manuscript - which is important but the negative tone risks "blaming" the veterinary profession and doesn't recognise that the treatment of animals is also necessary to maintain animal welfare. The distribution of antimicrobial classes is pretty consistent with other studies and the high use of category C is not so concerning as the use of category B agents. I think the tone of the paper should be adjusted to be more circumspect and balanced.
Figures should have legends not titles.
Discussion
The discussion needs substantial rewording. It is disjointed.
Line 302: Use EMA not WHO
Line 311: Do you mean category B? More nuanced discussion around this is warranted.
Conclusion:
Line 372: instead of "between the years 2019-2020" you could use "in the study period" as you didn't look at 2 full years.
Line 375: Incomplete sentence
Line 376-377: You didn't discuss guidelines at all in results or discussion. Please remove from conclusion
Line 377-379: You didn't do any analysis of diagnostics or culture and antibiograms so this is unfounded
Needs substantial English language editing
